# Rapid Self-Assembly of Polymer Nanoparticles for Synergistic Codelivery of Paclitaxel and Lapatinib via Flash NanoPrecipitation

**DOI:** 10.3390/nano10030561

**Published:** 2020-03-20

**Authors:** Shani L. Levit, Hu Yang, Christina Tang

**Affiliations:** 1Chemical and Life Science Engineering Department, Virginia Commonwealth University, Richmond, VA 23284, USA; levitsl@vcu.edu (S.L.L.); hyang2@vcu.edu (H.Y.); 2Department of Pharmaceutics, Virginia Commonwealth University, Richmond, VA 23298, USA; 3Massey Cancer Center, Virginia Commonwealth University, Richmond, VA 23298, USA

**Keywords:** Flash NanoPrecipitation, polymer nanoparticle, codelivery, combination therapy, drug synergy, ovarian cancer, nanomedicine

## Abstract

Taxol, a formulation of paclitaxel (PTX), is one of the most widely used anticancer drugs, particularly for treating recurring ovarian carcinomas following surgery. Clinically, PTX is used in combination with other drugs such as lapatinib (LAP) to increase treatment efficacy. Delivering drug combinations with nanoparticles has the potential to improve chemotherapy outcomes. In this study, we use Flash NanoPrecipitation, a rapid, scalable process to encapsulate weakly hydrophobic drugs (logP < 6) PTX and LAP into polymer nanoparticles with a coordination complex of tannic acid and iron formed during the mixing process. We determine the formulation parameters required to achieve uniform nanoparticles and evaluate the drug release in vitro. The size of the resulting nanoparticles was stable at pH 7.4, facilitating sustained drug release via first-order Fickian diffusion. Encapsulating either PTX or LAP into nanoparticles increases drug potency (as indicated by the decrease in IC-50 concentration); we observe a 1500-fold increase in PTX potency and a six-fold increase in LAP potency. When PTX and LAP are co-loaded in the same nanoparticle, they have a synergistic effect that is greater than treating with two single-drug-loaded nanoparticles as the combination index is 0.23 compared to 0.40, respectively.

## 1. Introduction

Ovarian cancer remains one of the most difficult cancers to treat due to late-stage diagnosis and its highly malignant nature [1]. Chemotherapies such as Taxol, a formulation of paclitaxel (PTX), remains to be one of the most widely used cancer treatments, particularly for recurring ovarian carcinomas following surgery [1,2,3]. The mechanism of action for PTX is binding to the β-subunit of tubulin at two sites, which stabilizes the tubulin polymers preventing cytoskeletal rearrangement for cellular function [4,5,6]. Stabilizing tubulin results in cell cycle arrests in the G_2_/M phase [6]. However, there are many challenges with the use of Taxol. There are severe systemic side effects associated with PTX treatment such as low blood pressure, risk of infection, the formation of blood clots, and neurotoxicity [7,8,9]. Additionally, PTX is poorly water-soluble and has low permeability which limits drug efficacy due to low drug concentrations reaching the tumor site [10]. Clinically, PTX is used in combination with other drugs to increase the efficacy of treatment by targeting multiple pathways [11,12,13]. 

Paclitaxel is often used in combination with lapatinib (Tykerb, LAP), a class of tyrosine kinase inhibitors [14,15,16,17]. Several studies observed an increase in drug efficacy in terms of tumor cell death and decrease in tumor volume when PTX and LAP were used in combination [15,18]; in some cases, a synergetic effect was observed [19]. However, combination treatment required premedication before injection, i.e., complex treatment regimens with multiple methods of administration [20,21]. Formulation of drug combinations in nanoparticles could overcome low solubility and permeability of the drugs to deliver an effective drug dosage to the tumor site and simplify drug regimens to improve patient adherence, while decreasing side effects [7,22,23]. 

Co-encapsulation of PTX and LAP may improve the co-localization of the drugs in the tumor tissue and increase drug efficacy [7,24,25,26,27]. For example, PTX and LAP have been co-formulated in a core-shell structure using polymer micelles. Lapatinib was conjugated to a PEGylated block copolymer and formulated into micelles encapsulating PTX in the core. Interestingly, formulation into the polymer micelles increased the potency of the PTX as indicated by a two-fold decrease in the half-maximal inhibitory concentration (IC-50) concentration in certain types of breast cancer [25]. Although the increase in potency via formulation into nanoparticles is exciting, this approach requires multiple steps and covalent modifications of LAP which results in the formation of a new compound, requiring further testing for FDA approval, a costly and time-consuming process [28]. 

Nanoparticle formulations co-encapsulating PTX and LAP without the chemical modification of LAP have also been considered [7,24,25,26,27,29]. Vergara et al. co-encapsulated LAP and PTX in polyelectrolyte nanoparticles by the sonication-assisted layer-by-layer (SLBL) technique. To formulate these nanoparticles, PTX-chitosan nanoparticles were first formed, followed by the sequential deposition of alginic acid and chitosan coatings. Lapatinib was co-deposited with chitosan to achieve nanoparticles with a PTX core and LAP shell. The core-shell nanoparticles showed a significant decrease in cell viability in vitro compared to PTX-loaded nanoparticles and free PTX [7]. While the results are promising, the formulation of the nanoparticles was time intensive as each deposition of each layer required 20–45 min. 

Co-loading both LAP and PTX in the nanoparticle core has been achieved using lipopolymer [24] or Pluronic F127 polymeric micelles [30]. Formulation using the Pluronic F127 suppressed tumor cell proliferation and decreased IC-50 by 10-folds relative to the free drug combination treatment of PTX and LAP [30]. These nanoparticles provide the basis for further improvements of drug combinations; however, the formulation method is challenging to scale up [31]. Furthermore, the drug effect when co-delivering drugs in nanoparticle form in terms of synergy is not well established. 

In this study, we extend the use of Flash NanoPrecipitation (FNP) to PTX and LAP by leveraging in situ coordination complexation of tannic acid and iron. Flash NanoPrecipitation enables the rapid, scalable formulation of drug combinations [32]. However, this method has generally been limited to highly hydrophobic materials (logP > 6) [33]. Encapsulating PTX and LAP using FNP is challenging due to their relatively weak hydrophobicity (PTX, logP = 3.2 and LAP, logP = 5.4). We encapsulate drugs (logP < 6) via in situ coordination complex formation of tannic acid–iron (TA–Fe) and stabilization with an amphiphilic block copolymer. Our focus is on understanding how incorporating multiple drugs affects nanoparticle self-assembly. Based on our understanding, we establish the formulation parameters to form PTX NPs, LAP NPs, and PTX–LAP NPs with comparable sizes (~100 nm in diameter). We perform initial drug release studies in vitro, focusing on the release of PTX. We evaluate the potency of the nanoparticles in vitro using an ovarian cancer cell line OVCA-432. The core of our preliminary in vitro evaluation is based on IC-50 values; the effect of co-encapsulating the drugs in terms of synergy using the combination index is analyzed. 

## 2. Materials and Methods

### 2.1. Materials

HPLC grade tetrahydrofuran (THF), dimethyl sulfoxide (DMSO), acetonitrile, and Tween 80 were purchased from Fisher Scientific (Pittsburg, PA, USA). ACS grade tannic acid (TA) and ACS grade iron (III) chloride hexahydrate (97%) were purchased from Sigma-Aldrich (St. Louis, MO, USA). Paclitaxel (PTX, >98%) and lapatinib (LAP, >98%) were obtained from Cayman Chemical Company (Ann Arbor, MI, USA); phosphate-buffered saline without calcium and magnesium was purchased from Lonza (Basel, Switzerland). Polystyrene-b-polyethylene glycol (1600-b-500 g/mol) (PS-b-PEG) was obtained from Polymer Source (Montreal, Quebec, Canada) and was purified by dissolving in THF (~40 °C) and precipitating into diethyl ether then dried by vacuum for two days as previously described [34].

### 2.2. Cell Culture

Ovarian cancer cell line OVCA-432 was a kind gift from Dr. Xianjun Fang from Virginia Commonwealth University. The OVCA-432 cells were cultured in RPMI-1640 media containing 2 mM L-glutamine (ATCC, Manassas, VA, USA) supplemented with 10% Fortified Bovine Calf Serum (FBS, HyClone Cosmic Calf Serum, Fisher Scientific, Pittsburg, PA, USA), 100 U/mL penicillin and 100 µg/mL streptomycin (Gemini Bio-Products, West Sacramento, CA, USA), and cultured at 37 °C at 5% CO_2_. The cells were passaged once a week.

### 2.3. Nanoparticle Formulation

Flash NanoPrecipitation (FNP) was used to prepare polymer-based nanoparticles encapsulating the anti-cancer drugs with a hand-operated confined impinging jet (CIJ) mixer with dilution as previously described [35,36]. Four nanoparticles were formulated that either encapsulated the TA–Fe complex (TA–Fe NPs), PTX (PTX NPs), LAP (LAP NPs), or both PTX and LAP (PTX–LAP NPs).

To self-assemble the nanoparticles, PS-b-PEG, TA (4 mg/mL), and one or more of the drugs (PTX and LAP) were dissolved in a water-miscible organic solvent (e.g., THF or DMSO) by sonication (~40 °C) for 10 min to formulate the organic stream. The organic stream was rapidly mixed with the Fe^3+^ (aq., 1 mg/mL) at equal volumes, typically 1 mL, in the CIJ mixer. The effluent from the mixer was immediately diluted in 1X PBS at pH 7.4 for a final organic solvent/water ratio of 1:9 by volume. The drug concentration in the organic stream of PTX and LAP was varied from 0.5 mg/mL to 2 mg/mL; the block copolymer concentration was varied relative to the core material. Specifically, the core material was considered the TA and Fe^3+^ for the TA–Fe NPs, and for the drug-loaded nanoparticles it was determined as TA and the drugs encapsulated. The ratio of the PS-b-PEG to the core material was varied between 1:1 and 2:1 by mass.

Within 24 h of formulation, the nanoparticles were filtered to remove the organic solvent, unencapsulated drug(s), and excess TA and Fe^3+^ with Amicon Ultra-2 Centrifugal filters (Amicon Ultracentrifuge filter (Ultracel 50K, 50,000 NMWL), Merck Millipore Ltd., Burlington, MA, USA) by centrifuging at 3700× *g* rpm for ~15–30 min (5804 R 15 amp version, Eppendorf, Hamburg, Germany). The nanoparticle pellet was resuspended with 1X PBS to a nominal concentration of ~25 mg/mL of total solids and stored at ~4 °C. The nanoparticles were used within 5 days of the FNP to ensure there was minimal change in particle size and drug loss. 

### 2.4. Nanoparticle Characterization

The size, polydispersity (PDI), and zeta potential of the nanoparticles were characterized immediately after FNP and after filtration using dynamic light scattering (Malvern Zetasizer ZS, Malvern Instruments Ltd., Malvern, United Kingdom). The nanoparticle size and polydispersity index (PDI), a measure of uniformity, were measured by averaging 4 measurements at a scattering angle of 173°. Nanoparticles populations with a PDI of less than 0.400 were considered uniform [37]. The nanoparticle size stability at 4 °C was observed by measuring size and PDI for up to 3 weeks after formulation. The concentration of the nanoparticle dispersion following filtration was determined by thermogravimetric analysis (TGA) (Pyris 1 TGA, Perkin Elmer, Waltham, MA, USA).

Transmission electron microscopy (TEM) samples were prepared by diluting the filtered nanoparticle dispersions with DI water 1:20 by volume ratio and pipetting 5 μL three times onto a TEM grid with Formvar/Carbon support films (200 mesh, Cu, Ted Pella, Inc., Redding, CA, USA) and dried under ambient conditions. Dilution was necessary to prevent aggregation during drying. The samples were imaged with a JEOL JEM-1230 (JEOL USA, Inc., Peabody, MA, USA) at 120 kV. 

To determine the drug content of the nanoparticles, acetonitrile (1.8 mL) was added to nanoparticles (50 μL) filtered with Amicon filter, as previously described, and the sample was vortexed so that the nanoparticles would disassemble. The sample was centrifuged at 10,000× *g* rpm for 6 min, and then the supernatant was collected for reverse-phase high-performance liquid chromatography (RP–HPLC) (1260 HPLC with Quaternary Pump and UV–Vis Diode Array Detector, Agilent, Santa Clara, CA, USA) fitted with a Luna^®^ 5 µm C18 100 Å, LC Column 250 × 4.6 mm (Phenomenex, Torrance, CA, USA). The sample was eluted with degassed acetonitrile and water gradient at a flow rate of 1 mL/min (0–1 min at 80:20, 1–6 of ramp up to 0:100, 6–8 min at 0:100, and ramp down to 80:20 between 8 and 9 min). PTX was measured at a wavelength of 228 nm with a retention time of ~8 min and LAP was measured at 332 nm with a retention time of ~9 min. The concentration of each drug was determined by comparing the peak areas with the standard calibration curve. Encapsulation efficiency (EE%) and drug loading (DL%) were calculated based on Equations (1) and (2), respectively, and the values reported are the average and standard deviation of three trials.
(1)$$Encapsulation\;efficiency\;\left(EE\%\right)=\frac{Mass\;of\;drug\;encapsulated}{Initial\;mass\;of\;drug}\times 100\%$$
(2)$$Drug\;loading\;\left(DL\%\right)=\frac{Mass\;of\;drug\;encapsulated}{Total\;nanoparticle\;mass}\times 100\%$$

### 2.5. Nanoparticle Drug Release In Vitro

To measure the drug release, 500 μL of concentrated nanoparticle dispersion was loaded into a 7000 MWCO dialysis unit (Slide-A-Lyzer_®_ MINI Dialysis Unit, Thermo Scientific, Waltham, MA, USA) and incubated in 0.5% Tween 80 in PBS at pH 7.4 at 37 °C, which was replaced every day of the experiment. Samples (32 μL) at 0 h, 3 h, 6 h, 24 h, 48 h, day 4, day 6, and day 10 were taken from the nanoparticle dispersion and the remaining drug concentration was determined by RP–HPLC as previously described. Three replicates of each drug-loaded nanoparticle dispersion were tested.

### 2.6. Cytotoxicity

OVA-432 cells were seeded at a density of 15 × 10^3^ cell/well in a 96-well plate containing 100 µL of complete medium. The cells were incubated at 37 °C in 5% CO_2_ overnight. Then the media was replaced with 100 µL medium containing free drugs or nanoparticles and treated for 48 h. Stock solutions of free drug were prepared by dissolving PTX (12 mg/mL) or LAP (5 mg/mL) in DMSO and sonicating for 5 min. Then, the drugs were diluted with complete media and serial dilutions were performed to achieve concentrations between 200 and 0.0002 µg/mL. Additional DMSO was added for a final DMSO concentration of 2% *v/v*. The nanoparticles were concentrated with Amicon filters (50 kDa MWCO) as previously described and the nanoparticle pellet was diluted with 1X PBS. The PTX NPs and LAP NPs were individually concentrated to 1,000 μg/mL of drug. The PTX–LAP NPs were concentrated to 500 µg/mL relative to PTX. The nanoparticle-loaded medium was prepared by diluting the stock nanoparticle dispersion with complete media and performing serial dilutions for final concentrations between 200 and 0.0002 µg/mL. The cells were also treated with complete media and 2% *v/v* DMSO media as controls for comparison. There were 6 replicates for each experimental condition. After 48 h, the cell viability was measured with WST-1 assay (Sigma-Aldrich, St. Louis, MO, USA) according to manufacturing instructions. Briefly, the drug-loaded medium was removed and 100 µL of RPMI-1640 with Phenol Red (Fisher Scientific, Pittsburg, PA, USA) containing 10% WST-1 solution was added to each well as well as to 6 empty wells. The cells were incubated between 45 and 90 min until there was a visible color change to a golden-yellow or the absorbance of control wells reached at least 0.700 measured with a microplate reader (VersaMax ELISA microplate reader, Molecular Devices, San Jose, CA, USA) at a wavelength of 440 nm with background subtraction of 640 nm. The cell viability was determined by subtracting the background noise (wells containing only 10% WST-1 in media) from the samples and then dividing the sample absorbance by the average absorbance of the untreated wells. The relative cell viability was expressed as a percentage of the untreated cells with a mean ± standard deviation of six replicates.

### 2.7. Cell Cycle Analysis by Flow Cytometry

The cells were seeded at a density of 20 × 10^4^ cells/mL in a 35 mm petri dish containing 3 mL of complete media. The cells were incubated at 37 °C and 5% CO_2_ until 90% confluence and the media was replaced every 2 days. The cells were treated with either free PTX, free LAP, PTX NPs, LAP NPs at the IC-50 concentration, or left untreated for 48 h at 37 °C. After 48 h treatment, the cells were stained with Propidium Iodide (PI Flow Cytometry Kit, Abcam, Cambridge, MA, USA) for flow cytometry according to the manufacturer’s instructions. Briefly, the cells were trypsinized and the aspirated medium and PBS were collected to minimize cell loss. The cells were centrifuged at 700× *g* for 5 min as necessary. The cells were washed with 1X PBS and fixed with 66% ethanol by slowly adding ethanol to PBS during vortexing. The cells were stored in ethanol at 4 °C for at least 2 h and up to 4 days. The cells were centrifuged and washed with PBS to remove the ethanol. The 1× Propidium Iodide and RNase solution was prepared immediately prior to use by mixing 5% *v/v* of 20× Propidium Iodide and 0.05% *v/v* 200X RNase in 1X PBS. Then the cells were resuspended in 200 µL/500,000 cells of 1X Propidium Iodide and RNase solution and incubated in the dark at 37 °C for 30 min. Prior to flow cytometry, the cell samples were stored on ice and filtered through a cell strainer (Falcon Test Tube with Snap Cap, Fisher Scientific, Pittsburg, PA, USA). Flow cytometry was performed on a BD FACSCanto™ II Analyzer (BD Biosciences, San Diego, CA, USA) and 10,000 cells were analyzed at an excitation of 488 nm and emission of 670 nm. The samples were analyzed in triplicate.

## 3. Results and Discussion

### 3.1. Nanoparticle Preparation and Characterization

Flash NanoPrecipitation (FNP) is a well-established polymer-directed self-assembly method for preparing size-tunable nanoparticles encapsulating highly hydrophobic molecules (logP > 6). Since nanoparticle self-assembly involves adsorption of the hydrophobic block of the block copolymer to a precipitating core material, this process has generally been limited to highly hydrophobic materials with a logP of six or greater [32,33]. Since PTX is not sufficiently hydrophobic to form stable particles via FNP directly [38], we explore an alternative approach in which we encapsulate PTX (logP = 3.2) and LAP (logP = 5.4) using a pH-labile, tannic acid–iron (TA–Fe) based nanoparticle platform [35]. 

To prepare TA–Fe based nanoparticles, FNP was performed by mixing drug(s), dissolving TA and PS-b-PEG in a water-miscible (THF or DMSO) organic solvent with iron (III) chloride dissolved in water in a confined impinging jet mixer. The effluent from the mixer was quenched in a bath of PBS, pH 7.4, conditions under which the TA–Fe coordination complex is expected to be insoluble. Upon rapid mixing, the TA and Fe^3+^ form an insoluble coordinate complex which co-precipitates with the drug(s) forming the particle core. Precipitation of the core materials is arrested by adsorption of the hydrophobic block of the block copolymer and the PEGylated end of the block copolymer sterically stabilizes the nanoparticle in dispersion (Appendix A). The dispersions appeared red which is consistent with the tris-complex of TA and iron expected at pH 7.4 [31,39]. Nanoparticles encapsulating the TA–Fe complex (TA–Fe NPs) are 109 ± 5 nm (Appendix A) with a zeta potential of –21.4 ± 2.1 mV consistent with other PEGylated nanoparticles [33,40]. 

Our initial goal was to achieve uniform PTX-loaded nanoparticles on the order of 100 nm to allow for passive targeting [41]. We examined the effect of organic solvent selection, total solids concentration, ratio of the block copolymer to core materials, and drug concentration on the ability to make uniform particles and resulting nanoparticle size. 

Two water-miscible organic solvents were considered, THF and DMSO, as the block copolymer; TA, LAP, and PTX were sufficiently soluble for the self-assembly of nanoparticles via FNP. However, when DMSO was used for the organic stream, a visible red-purple precipitate formed immediately upon mixing in the reservoir. With THF, stable PTX-loaded nanoparticle dispersions were achieved with a zeta potential of –22.1 ± 2.1 mV and no precipitate was observed. These results suggest that co-precipitation of PTX and the TA–Fe complex and the rate of PS-b-PEG self-assembly is more appropriately matched using THF as a solvent than DMSO as a solvent. Thus, THF was used as a solvent in all further experiments.

To further tune the size of the PTX-loaded particles, we examined the effect of other formulation parameters. At a total solid concentration of 18 mg/mL in the streams and above, there are two size populations in the intensity weighted distribution with peak diameters of ~30 nm and ~100 nm. The population with a hydrodynamic diameter of ~30 nm can be attributed to empty block copolymer micelles [35,42] produced with the PTX-loaded nanoparticles of ~100 nm (Table 1). The inability to form uniform particles at high concentrations has been previously observed and could be attributed to a limited affinity between stabilizer and TA–Fe precipitates at high iron concentrations [35]. 

To improve particle uniformity, we next examined the ratio of the block copolymer to core materials (BCP: core with the core defined as a concentration of TA and PTX in the formulation) at reduced total solids concentration. Specifically, three BCP: core ratios, i.e., 1:1 1.5:1, and 2:1, were studied with a total solid concentration of less than 16 mg/mL. All three ratios produced uniform nanoparticles with a PDI < 0.400. At a 1:1 BCP: core ratio, the particles were 170 ± 33 nm. Increasing the amount of block copolymer from a 1:1 to 2:1 BCP: core ratio produced a 35% decrease in particle size (Table 1) and, for the 2:1 BCP: core ratio, uniform PTX-loaded particles with a hydrodynamic diameter of 117 ± 3 nm were achieved. TEM confirmed the nanoparticles were spherical and the size was consistent with DLS (Appendix A). This trend has been attributed to an increase in the rate of self-assembly relative to the rate of core growth, limiting the growth of the core before it is kinetically stabilized. These results are comparable to FNP systems using hydrophobic core materials (logP > 6) in which the particle size can be tuned by varying the ratio of the block copolymer to the core [32,39].

Thus, we next investigated the effect of PTX concentration to maximize the drug loading in the nanoparticle while maintaining uniform, ~100 nm nanoparticle formulations (Figure 1). We varied the PTX concentration from 0.5 to 2 mg/mL in the organic stream. Increasing the PTX concentration to 2 mg/mL resulted in two size populations with peak diameters of ~100 nm and ~20 nm (Table 1) possibly due to a mismatch time scales of complexation/precipitation and block copolymer micellization at higher concentrations of the drug. The highest concentration that we used that resulted in uniform PTX-loaded particles was 1 mg/mL. 

In parallel with formulating PTX-loaded nanoparticles, we also used FNP to encapsulate LAP within TA–Fe nanoparticles via in situ complexation with the aim of achieving uniform LAP nanoparticles ~100 nm in diameter. 

Similar to PTX, total solids concentration above 36 mg/mL in the streams resulted in two size populations in the intensity weighted distribution with peak diameter ~30 nm and ~100 nm due to the formation of empty block copolymer micelles. Lowering the total solids concentration to 16 mg/mL, nanoparticle dispersions with uniform size were achieved (Table 2). Nanoparticle dispersions with uniform particle size were obtained at BCP to core ratios between 2:1 and 1:1. Interestingly, the size of the LAP-loaded NPs was independent of BCP: core ratio. This result indicates that the concentrations used in the rate of LAP/TA–Fe co-precipitation is comparable to the self-assembly of PS-b-PEG micellization.

Next, we investigated the effect of LAP concentration to maximize the drug loading in the nanoparticle while maintaining uniform size distributions (diameter ~100 nm). At LAP concentrations 2 mg/mL nanoparticles were produced at ~150 nm and ~30 nm (Appendix A). These results are comparable to the results observed with PTX. Reducing the drug concentration, uniform, ~100 nm particles were achieved and confirmed with TEM imaging (Figure 1 and Appendix A). These results suggest co-precipitation of these weakly hydrophobicity drugs (logP < 6) with the TA–Fe core affects the timescale of precipitation as well as the affinity of the stabilizer and the core that can result in the formation of empty micelles and need to be considered when formulating these drug-loaded nanoparticles. 

Finally, our goal was to produce co-loaded nanoparticles containing both PTX and LAP (PTX–LAP NPs). Based on the findings from formulating PTX NPs and LAP NPs, we first focused on drug concentration using THF as the organic solvent. Using a drug concentration of 1 mg/mL PTX and 1 mg/mL LAP (a total drug concentration of 2 mg/mL), nanoparticles with multiple two size populations (peak diameters of 119 ± 28 nm and 22 ± 3 nm) were observed. When the total drug concentration was decreased to 1 mg/mL (0.5 mg/mL each of PTX and LAP), uniform nanoparticles were produced at 115 ± 3 nm (Appendix A). These results are comparable with PTX NPs and LAP NPs where a maximum drug concentration of 1 mg/mL could be used in the formation of monodispersed nanoparticles. To maximize the drug loading, a BCP: core ratio of 1:5:1 was used. TEM confirms the particles are spherical and the particle size is consistent with DLS (Figure 1 and Appendix A). Additionally, the nanoparticle size and polydispersity were unaffected by the filtration process (Appendix A). Nanoparticle size was stable for up to two weeks after FNP when stored at 4 °C (Appendix A).

Following FNP, the drug concentration in the resulting dispersions was determined by HPLC after disassembling the nanoparticles with acetonitrile. From the drug concentration, the encapsulation efficiency and drug loading of PTX and LAP were determined. The encapsulation efficiency of the drug is the amount of drug encapsulated compared to the nominal amount in the formulation. The drug loading of PTX and LAP in the single-drug-loaded nanoparticles were similar (Table 3) and comparable to previous literature using polymer micelles [27,43]. For the single-drug-loaded nanoparticles, the encapsulation efficiency PTX and the LAP were 37.6% ± 14.4% and 25.0% ± 1.5%, respectively, which are comparable to previous reports using polymer micelles [43]. Interestingly, in the co-loaded nanoparticles, the encapsulation efficiency of PTX increases from 37.6% ± 14.4% to 67.0% ± 2.2% while the encapsulation efficiency for LAP in PTX–LAP NPs remained the same as the LAP NPs (Table 3). This result could be attributed to a more hydrophobic core environment in the presence of LAP facilitating encapsulation of PTX. Due to the selective increase in encapsulation efficiency of PTX in the presence of LAP, the drug loading of PTX was 2.7-fold higher than the LAP loading (2.11% ± 0.50% compared to 0.79% ± 0.49%) despite using equal amounts of each drug during mixing (0.5 mg/mL of each). Notably, these drug concentrations are comparable to previous studies micelles [27,43]. 

### 3.2. Drug Release

As a first step to understanding the in vitro drug release rates of PTX and LAP from nanoparticles, dialysis was performed with PBS at pH 7.4 with Tween 80 similar to previous studies [44,45,46,47]. Examining the PTX-loaded nanoparticles, a burst release was observed within the first six hours at which ~20% of PTX was released. After six hours, the burst release was followed by sustained PTX release and the drug release plateaued at ~40% on day six (Figure 2A). In comparison, ~25% of LAP released from LAP-loaded NPs in the first three hours (~25%). Following the burst release, the sustained release of LAP release over six days was observed with ~35% total drug release achieved (Figure 2B). 

PTX release from the co-loaded nanoparticles was comparable to the single-drug-loaded nanoparticles with burst release in the first six hours and cumulative drug release at day six of ~40%. We examined the drug release kinetics of PTX from single-drug and co-loaded nanoparticles and fitted it to the Korsmeyer–Peppas diffusion model (Equation (3)) [48,49]
(3)$$\frac{M_{t}}{M_{\infty }}=at^{n}$$
where the *M_t_* is the drug release at time, *t*, *M_∞_* is maximum drug release, and *a* is the release rate. The diffusion exponent, *n*, is determined based on the fit and described the drug release mechanism [48]. The diffusion exponent for PTX released from PTX NPs and from PTX–LAP NPs was 0.34. Since the diffusion exponent was less than 0.45, it indicates first-order Fickian diffusion kinetics [50,51]. 

Examining the LAP release, we observe a slight decrease in cumulative release after 24 h (Figure 2B). The fluctuations for lapatinib release from the nanoparticles are unusual but similar observations have been previously reported in other drug release systems [52,53]. The fluctuations in cumulative release can be potentially attributed to the supersaturation of lapatinib in the dialysis media in the first 24 h due to the burst release of the drugs from the nanoparticles. Supersaturation could cause nanoprecipitation of LAP which could result in the apparent drop in cumulative drug accumulation [54]. This phenomenon has been observed with other hydrophobic drugs from nanoparticles [52,54].

Investigating release from the PTX–LAP NPs, there is a decrease in cumulative release in LAP between 6 and 24 h (Figure 2C). This result suggests that the release of PTX increases the supersaturation of and nanoprecipitation of LAP. Notably, when comparing LAP release from the co-loaded nanoparticles to the single-drug-loaded nanoparticles, burst release occurred over six days rather than three days and the cumulative LAP release at six days was two-folds lower for co-loaded nanoparticles compared to the single-drug nanoparticle (~16% compared to ~35%) (Figure 2C). Co-encapsulating PTX and LAP into nanoparticles resulted in a decrease in the cumulative drug release of LAP but the drug release was comparable for PTX to single-drug-loaded nanoparticles. The slower burst release of LAP from co-loaded nanoparticles may be attributed to lower drug loading concentrations compared to LAP NPs resulting in a slower dissolution profile, a phenomenon observed with hydrophobic materials [55]. These results are consistent with previous literature indicating LAP has a slower release profile compared to PTX from polymer micelles [30]. Studies to further characterize the drug release, especially LAP, using alternative media (e.g., other surfactants, or biologically relevant media such as full growth medium with serum) will be pursued in future work.

### 3.3. Assessing Drug Efficacy of Single-Drug Nanoparticles

Finally, the efficacy of the nanoparticle dispersions was assessed in vitro with ovarian cancer cells, OVCA-432. Specifically, we used the IC-50 concentration, i.e., the drug concentration that reduces the cell viability by 50%, as a measure of potency. As a control, the cell viability was first examined for cells treated with TA–Fe nanoparticles without drugs. When treated with 50 µg/mL based on total solids concentration the cell viability was 95% (Appendix A). Examining the dose–response curve, the IC-50 concentration for the TA–Fe nanoparticles was ~1000 µg/mL of total solids concentration. This result demonstrates that the nanoparticle platform itself has minimal cytotoxic effects consistent with previous reports [35].

Next, we compared the potency of the nanoparticles compared to the free drug at the equivalent free drug concentration. We note at the concentrations of nanoparticles used, the (TA–Fe NPs) alone did not induce cytotoxic effects and the IC-50 was reproducible with OVCA-432 cells (Appendix A). Encapsulating the PTX shifts the dose–response curve to lower concentrations compared to free PTX (Appendix A) indicating an increase in potency upon encapsulation. A similar trend was observed for LAP (Appendix A). Interestingly, the dose–response curve of PTX NPs and LAP NPs plateaued at ~20% cell viability. At low drug concentrations, the TA in the nanoparticle could counter the effects of the drugs by inducing an antioxidant effect and eliminate free radicals produced with the anticancer drugs [56,57].

Specifically, the IC-50 concentration decreases from 70.6 ± 5.1 µg/mL for free PTX to 0.040 ± 0.003 µg/mL when encapsulated (*p* < 0.05) (Table 4). A similar result was observed for LAP; upon encapsulation, there was a nearly six-fold increase in potency as the IC-50 decreased from 4.6 ± 1.3 µg/mL for the free drug to 0.80 ± 0.26 µg/mL when formulated into nanoparticles (*p* < 0.05) (Table 4). While decreases in IC-50 concentration compared to the free drug form have been observed in other polymer nanoparticle formulations [24,58,59] and are not fully understood, the 1500-fold increase in PTX potency in this nanoparticle is noteworthy. The significant increase in PTX potency in the TA–Fe could be attributed to several contributing factors including sustained release over the 48-hour treatment period and increased bioavailability due to the nanoparticle formulation [24,60,61]. 

### 3.4. Cell Cycle Analysis

To better understand the increase in drug potency upon encapsulation, we examined the effect of treatment on the cell cycle using flow cytometry. The difficulty of treating cancer is the rapid proliferation of tumor cells and the propensity for metastasis. It is vital that cancer treatments such as PTX inhibit proliferation. For example, PTX arrests cells in the G_2_/M phase by stabilizing microtubules and preventing their disassembly necessary for cell division [62]. Thus, we examined the effect of the nanoparticles on the cell cycle using flow cytometry. Specifically, we compared the cell cycle of cells treated with free PTX and PTX NPs at their respective IC-50 concentrations. The untreated control cells were primarily in the G_0_/G_1_ phase with only 9% in the G_2_/M phase. With free PTX, the percentage of cells in the G_0_/G_1_ phase drops from 62% to 45% and there is an increase in the number of cells in the G_2_/M phase to 31% (Figure 3A). These results indicate that free PTX formulations accumulate OVCA-432 cells in the G_2_/M phase and likely decrease the cell viability by preventing progression to mitosis [62]. LAP and LAP-NP-treated cells remained primarily in the G_0_/G_1_ phase (Figure 3B) and the proportions for LAP and LAP NPs were comparable. Thus, free LAP and LAP NPs seem to stabilize the cells in the G_0_/G_1_ phase over the 48-hour treatment with minimal progression to the subG_1_ phase as expected since LAP is known to arrest cancer cells in the G_1_ phase of the cell cycle [63]. 

While the control nanoparticles had no effect on the cell cycle (Appendix A), when OVCA-432 cells were treated with PTX NPs the proportion of cells in the G_0_/G_1_ phase is similar to free PTX treated cells. Notably, treatment with PTX NPs shifted the cells to the subG_1_ phase relative to both free PTX and control (Figure 3A) and decreased the proportion of cells in the G_2_/M phase from 31% to 11%. Increase proportion in the subG_1_ phase could indicate that cells spend a shorter time in the G_2_/M phase with rapid DNA fragmentation [64] or it could be attributed to a short period of G_1_ arrest followed by progression to the subG_1_ phase during the 48-hour treatment [65]. Importantly, cells in the subG_1_ phase undergo DNA damage, which can lead to cell death over time [66]. Overall, these changes in the cell cycle support our findings that encapsulation increases PTX potency compared to free PTX. 

### 3.5. Drug Combination and Synergy

Next, we examined the efficacy of the co-loaded formulation. Co-encapsulating the PTX with LAP further shifted the dose–response curve to lower concentrations compared to free PTX (Figure 4). Co-encapsulating PTX and LAP further increases PTX potency as indicated by the two-fold decrease in IC-50. 

Based on the IC-50 of PTX–LAP NPs, we compared the cell viability of OVCA-432 cells treated with a single-drug nanoparticle (0.009 µg/mL PTX or 0.004 µg/mL LAP) to co-loaded PTX–LAP NPs (Table 5). The control nanoparticles containing no drug (TA–Fe NPs) at the same total solid concentration (~0.5 µg/mL) exhibited no cytotoxic effects on the OVCA-432 cells. When the OVCA-432 cells were treated with PTX NPs, there was a slight decrease in the cell viability to ~80% whereas LAP NPs did not significantly affect cell viability. As expected, the PTX–LAP NPs reduced cell viability to ~50% which was significantly lower compared to both PTX NPs (*p* = 0.0002) and LAP NPs (*p* = 0.0001) (Figure 5). These results indicate that at equivalent drug concentrations, co-loaded PTX–LAP NPs had the greatest potency. 

Building on these results, we compared the combination treatment to the single-drug treatment to determine if there was a synergistic effect of co-treating the cells with PTX and LAP. Synergy was examined with the combination index (CI) based on the Chou–Talalay method and when the CI is below one, the combination treatment is synergistic compared to the sum of the individual drug treatments [67,68,69]. Free PTX and free LAP can be combined synergistically as indicated by the CI of 0.18. PTX and LAP target different mechanisms of the cell to produce an anticancer effect [70,71]. LAP inhibits the function of ABC transporters, which helps increase the intracellular concentration of PTX thereby increasing drug efficacy [24,72,73]. Similar synergistic effects were seen with chemotherapeutic agents and tyrosine kinase inhibitors [12,13].

Interestingly, the CI of the co-loaded PTX–LAP NPs was 0.23 comparable to the free drug combination and lower than co-delivered nanoparticles, which had a CI of 0.40 (Table 5). These results indicate an advantage to the co-encapsulation of both drugs within the same particle. The co-loaded nanoparticles could enhance the co-localization of the drugs particularly if the nanoparticles are endocytosed by tumor cells. Overall, we have presented a rapid and scalable approach to encapsulating chemotherapeutic combinations in a pH-labile nanoparticle platform that enhances the potency treatment of ovarian cancer in vitro.

## 4. Conclusions

In conclusion, we demonstrated the encapsulation of weakly hydrophobic drugs (logP < 6) into polymer nanoparticles in a rapid, scalable FNP process using in situ complexation of TA–Fe. The size of the resulting nanoparticles is stable at pH 7.4, facilitating sustained drug release via first-order Fickian diffusion. Importantly, the nanoparticles that encapsulate PTX and LAP are significantly more potent than the free drugs as indicated by the significantly decreased IC-50. Co-encapsulating PTX with LAP further increased potency. Additionally, co-encapsulating PTX and LAP had a synergistic effect compared to the free drug and greater than co-delivery of the single-drug-loaded nanoparticles indicating an advantage co-encapsulation of both drugs within the same particle. Building on this promising study, further studies to understand the cytotoxic effects (e.g., apoptosis), nanoparticle uptake and localization within cells, protein adsorption to nanoparticles, as well as evaluation in additional cell lines will be considered in future work. This is an intriguing approach for improving the potency of existing chemotherapeutics. 

## Figures and Tables

**Figure 1 nanomaterials-10-00561-f001:**
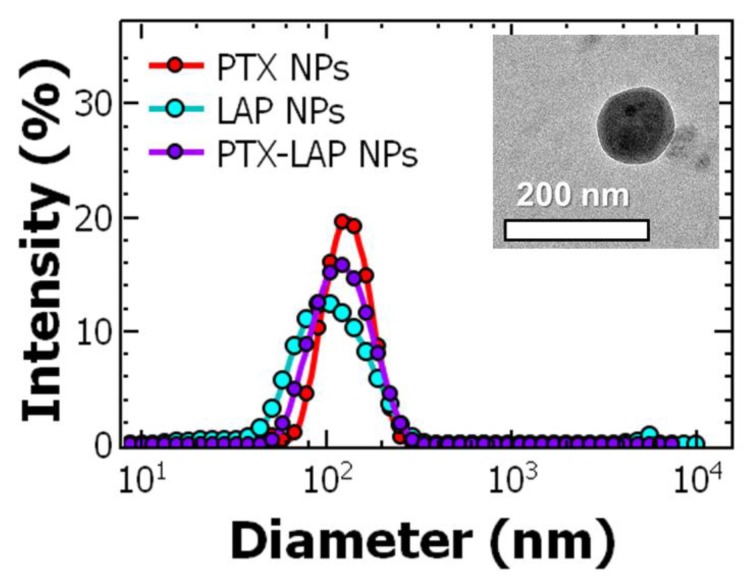
Representative dynamic light scattering results of uniform (red) paclitaxel nanoparticle (PTX NPs), (blue) lapatinib nanoparticle (LAP NPs), and (purple) co-loaded paclitaxel–lapatinib nanoparticle (PTX–LAP NPs) samples produced at ~100 nm (each DLS curve is the average of *n* = 4 measurements). Representative transmission electron microscopy (TEM) image of PTX–LAP NPs (scale bar = 200 nm) as inset.

**Figure 2 nanomaterials-10-00561-f002:**
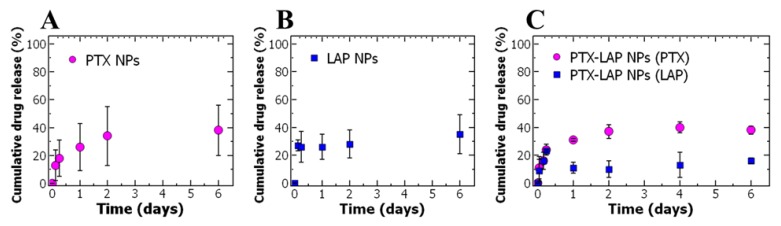
The cumulative drug release of paclitaxel (PTX) and lapatinib (LAP) from polymer nanoparticles (NPs) from (**A**) PTX from PTX NPs, (**B**) LAP from LAP NPs, and (**C**) PTX and LAP from co-loaded nanoparticles. The graph shows the average ± standard deviation of 3 replicates of FNP and independent drug release assays.

**Figure 3 nanomaterials-10-00561-f003:**
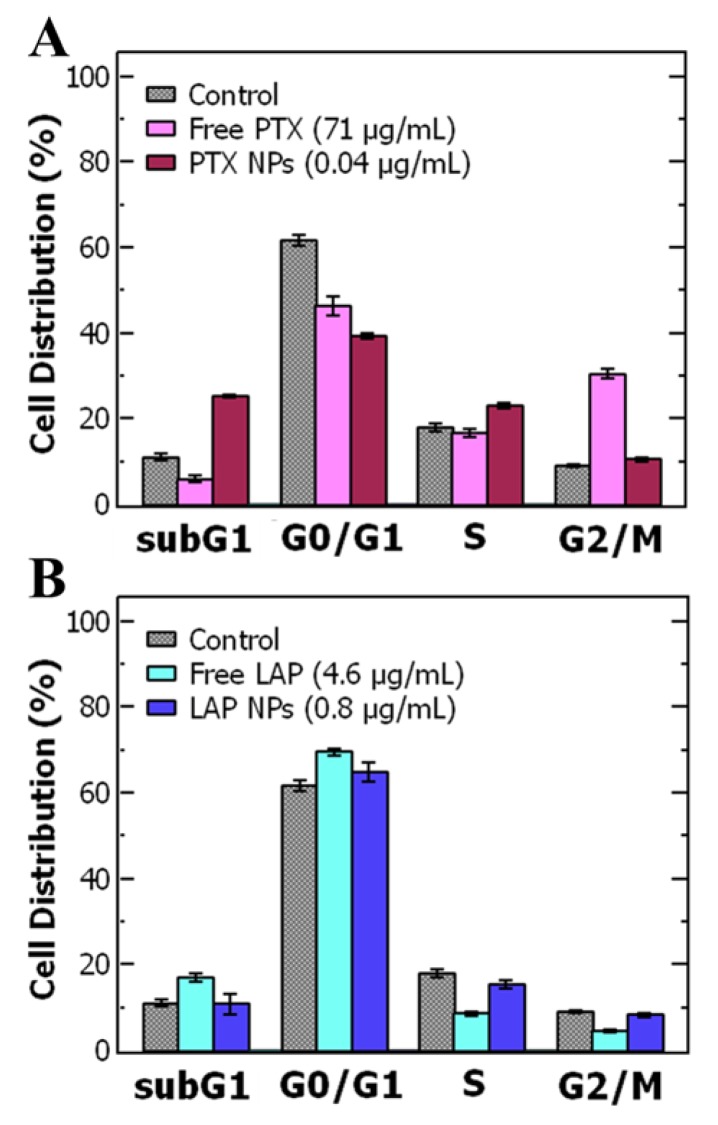
Cell cycle analysis of OVCA-432 cells from flow cytometry to compare free drug (in 2% *v/v* DMSO) and nanoparticle formulations for (**A**) paclitaxel (PTX) and (**B**) lapatinib (LAP). The graph shows the average ± standard deviation from 3 replicate wells.

**Figure 4 nanomaterials-10-00561-f004:**
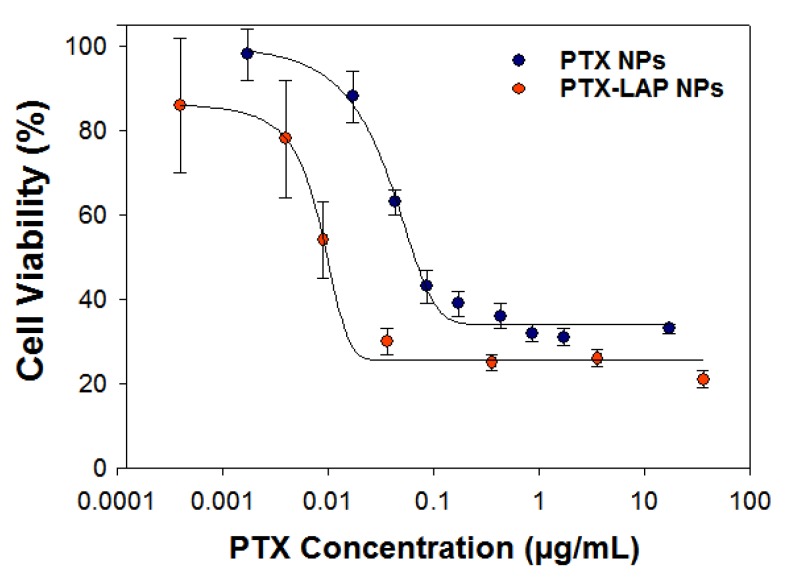
The cell viability dose–response curve of OVCA-432 cells when treated with (blue) paclitaxel nanoparticles (PTX NPs) and (red) paclitaxel–lapatinib nanoparticles (PTX–LAP NPs). The PTX–LAP NPs treatment shifts the dose–response curve to lower drug concentrations compared to the PTX NPs treatment. The graph shows the average ± standard deviation from one experiment performed with 6 replicate wells.

**Figure 5 nanomaterials-10-00561-f005:**
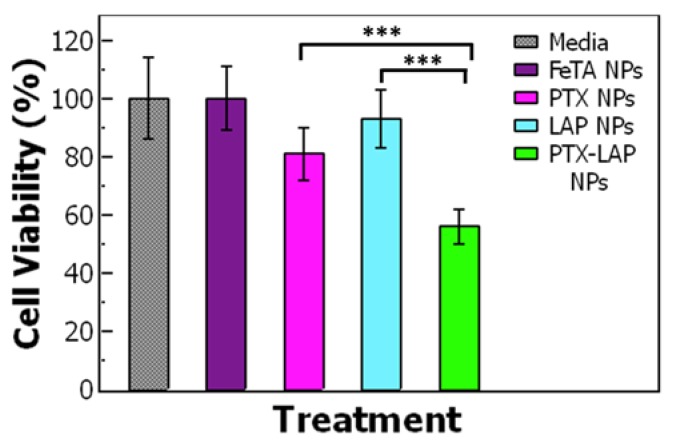
Cell viability of OVCA-432 cells after the 48-hour treatment with (gray) media, (purple) tannic acid–iron nanoparticles (TA–Fe NPs), (pink) paclitaxel nanoparticles (PTX NPs), (light blue) lapatinib nanoparticles (LAP NPs), (green) paclitaxel–lapatinib nanoparticles (PTX–LAP NPs). The cells were treated with a PTX concentration of 0.009 µg/mL and LAP at 0.004 µg/mL based on IC-50 of the PTX–LAP NPs. The cell viability was significantly lower when the cells were treated with PTX–LAP NPs when compared to PTX NPs (*p* < 0.05) and LAP NPs (*p* < 0.05). The graph shows the average ± standard deviation from one experiment performed with 6 replicate wells.

**Table 1 nanomaterials-10-00561-t001:** Summary of paclitaxel nanoparticle formulations.

Total Solids (mg/mL)	Ratio of BCP:Core	PTX Concentration (mg/mL)	Size 1 (nm) *	Size 2 (nm) *	PDI *
18	2:1	1	107 ± 2	32 ± 1	0.275 ± 0.009
36	2:1	2	113 ± 6	31 ± 1	0.372 ± 0.012
11	1:1	1	170 ± 33	0	0.142 ± 0.053
13.5	1.5:1	1	128 ± 7	0	0.244 ± 0.034
16	2:1	1	111 ± 10	0	0.255 ± 0.021
19	2:1	2	117 ± 3	20 ± 1	0.268 ± 0.009
16	2:1	1	111 ± 10	0	0.255 ± 0.021
14.5	2:1	0.5	134 ± 42	0	0.232 ± 0.145

* The average ± standard deviation of 3 replicates of Flash NanoPrecipitation (FNP) are reported.

**Table 2 nanomaterials-10-00561-t002:** Summary of lapatinib nanoparticle formulations.

Total Solids (mg/mL)	Ratio of BCP:Core	Size 1 (nm) *	Size 2 (nm) *	PDI *
16	2:1	91 ± 10	0	0.214 ± 0.038
36	2:1	106 ± 5	26 ± 4	0.288 ± 0.004
10.5	1:1	134 ± 8	0	0.255 ± 0.012
15.3	2:1	126 ± 7	0	0.380 ± 0.037

* The average ± standard deviation of 3 replicates of FNP are reported.

**Table 3 nanomaterials-10-00561-t003:** Encapsulation efficiency and drug loading of nanoparticles.

Samples	Encapsulation Efficiency (EE%) *		Drug Loading (DL%) *	
	PTX	LAP	PTX	LAP
PTX NPs	37.6 ± 14.4	-	3.11 ± 1.88	-
LAP NPs	-	25.0 ± 1.5	-	1.82 ± 0.71
PTX-LAP NPs	67.0 ± 2.2	25.9 ± 3.5	2.11 ± 0.50	0.79 ± 0.40

* The average ± standard deviation of 3 replicates of FNP are reported.

**Table 4 nanomaterials-10-00561-t004:** IC-50 of paclitaxel, paclitaxel nanoparticles, lapatinib, and lapatinib nanoparticles in OVCA-432 cells.

Drug Treatment	IC-50 (µg/mL) **	
	PTX	LAP
Free PTX *	70.6 ± 5.1	-
Free LAP *	-	4.6 ± 1.3
PTX NPs	0.040 ± 0.003	-
LAP NPs	-	0.80 ± 0.26

* In 2% DMSO with full growth medium. ** The average ± standard deviation (*n* = 6 treatments) are reported.

**Table 5 nanomaterials-10-00561-t005:** IC-50 and combination index of paclitaxel–lapatinib nanoparticles (PTX–LAP NPs) (co-loaded) compared to simultaneous delivery of paclitaxel nanoparticles (PTX NPs) and lapatinib nanoparticles (LAP NPs) (two single-drug-loaded NPs) in OVCA-432 cells.

Drug Treatment	IC-50 (µg/mL)		Combination Index (CI)
	PTX	LAP	
PTX NPs and LAP NPs	0.015 ± 0.003	0.007 ± 0.001	0.40
PTX-LAP NPs	0.0090 ± 0.0009	0.0040 ± 0.0004	0.23

The average ± standard deviation from one experiment performed with 6 replicate wells is reported.

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
