# Peer review of "Rapid Self-Assembly of Polymer Nanoparticles for Synergistic Codelivery of Paclitaxel and Lapatinib via Flash NanoPrecipitation"

_nanomaterials, 2020, doi:10.3390/nano10030561_

Round 1

Reviewer 1 Report

It is recommended that this paper can be accepted after a major revision.

  • The authors should do experiments of NPs cellular quantification.
  • The authors should do experiments of co-localization of NPs with intracellular organelles.
  • Please, add the analysis of protein corona.
  • It’s beneficial to test another cell line.

Reviewer 2 Report

Reviewer comments

In this manuscript the authors present data on a scalable method for combined incorporation of relevant drugs into nanoparticles. The topic is of general interest and has great potential. The manuscript is quite well written and the data are presented clearly. However, the manuscript in its present form has some weaknesses that have to be corrected before firm conclusions can be drawn.

1.The difference in toxicity between free drug and nanoparticle-carried drug is surprisingly large, particularly for PTX, which is also commented on by the authors: “The 1500-fold increase in PTX potency in this nanoparticle is noteworthy.” However, this finding may have several explanations that are not addressed in this manuscript:

  • This manuscript generally lacks information on how many independent experiments are performed for the viability after nanoparticle treatment and lacks statistics. In the Materials and Methods section it is stated: “There were 6 replicates for each experimental condition.” If the IC50 values listed in Table 4 is obtained from one cell culture experiment with 6 replicate wells for each condition, these experiments have to be repeated two more times and statistics have to be presented. Cell culture experiments are prone to variation, thus conclusions can never be drawn from just one experiment. The same particle batch can be used for both repetitions, as long as the cells are seeded on separate days, for instance two consecutive days. In this way the same particle batch can be employed within 5 days from synthesis.
  • From the description in Materials and Methods, it seems that the cell viability experiments for free drugs were performed in the presence of 2% DMSO, whereas the nanoparticle-loaded drugs were incubated with cells in the absence of DMSO. Firstly, 2% DMSO is approx 4x higher that what is recommended when performing cell culture experiments. DMSO at such high concentrations has been shown to have antioxidant effects by itself (Antioxidant properties of dimethyl sulfoxide and its viability as a solvent in the evaluation of neuroprotective antioxidants. Journal of pharmacological and toxicological methods 63(2):209-15, DOI: 10.1016/j.vascn.2010.10.004) or to alter cell cycle regulation and cell proliferation (Tunçer, S., Gurbanov, R., Sheraj, I. et al. Low dose dimethyl sulfoxide driven gross molecular changes have the potential to interfere with various cellular processes. Sci Rep 8, 14828 (2018). https://doi.org/10.1038/s41598-018-33234-z). Secondly, as the cells are incubated with 2% DMSO only in the free-drug condition and not in the presence of the nanoparticles, a direct comparison between the effect of free drug and nanoparticle-loaded drug is impossible. For a proper comparison, the cell culture experiments should be performed under identical conditions. Hydrophobic drugs will bind to serum proteins as a carrier in the complete cell culture medium, and thus a high concentration of DMSO should not be required to keep the drugs in suspension. The maximum dose of 0.5% DMSO, or preferably even less, should be used for free drug incubations.
  • In the comparison between free drug and drud-loaded nanoparticles, is the added concentration of nanoparticle-carried drug correlated to the expected release of nanoparticle-bound drug based on the in vitro drug release in PBS? The actual drug release in full cell culture medium may be highly different. In this connection, the measured drug release is described as: “Under biologically relevant conditions (PBS at pH 7.4) in vitro”. It would be more relevant to measure drug release in full growth medium with serum present, which more closely resembles blood. This would also visualize the release obtained in the cell culture experiments, which is the final readout for nanoparticle effects.
  1. TEM images: It is quite unusual to show just one particle. It would be more trustworthy to show numerous nanoparticles in one image and use the current images as enlarged inserts.
  2. In this manuscript the readout for cell viability is the WST-1 assay, which detects metabolic activity and not specifically cell death. When comparing the cytotoxicity of free drug and drug-loaded nanoparticles, all the nanoparticle preparations induce a reduction in WST-1 absorbance that reaches a plateau at high nanoparticle concentrations, whereas the free PTX reduces the metabolic activity further down. Have the authors checked that the nanoparticle-induced reduction in WST-1 absorbance is due to real cell death (as detected by plasma membrane permeabilization, for instance by PI exclusion), or could it be that the effect is merely cytostatic rather than cytotoxic? This issue is important in successful cancer treatment, and highly relevant in the initial studies of a potential new nanomedicine. As the authors discuss themselves, the tannic acid may act as an antioxidant and counteract the drug effect at higher nanoparticle concentrations.
  3. Figure 5 shows statistics, but there is no information on the number of independent experiments underlying the graphs and statistics. Such data have to be provided for each figure in the manuscript.

Reviewer 3 Report

  1. Please explain the decreasing cumulative release of the drug on figure 2/B at a third point and figure 2/c after the fourth point!
  2. Can you show TEM where more particles can be seen to check the uniformity of formed particles?
  3. In lines127-128 authors write that they have characterized the size distribution by DLS after  FNP and after filtration, what data have been given in the results. Please compare the size distribution determined after  FNP and after filtration.

Round 2

Reviewer 1 Report

  The paper can be published as it is.

Author Response

We thank the reviewer for all of their suggestions. The comments have helped clarify our manuscript and we are delighted that the reviewer considers our manuscript ready for publishing.

Reviewer 2 Report

I was surprised to learn that you regard 4-6 replicate wells as independent experiments. During 21 years of cell biology research, we have always performed at least three experiments on cells seeded on separate days. This can not be defined as studying the effect of cell age, but rather to study how robust your phenotype is to the inherent variations that occur during cell culture experiments. The cells in the 6 wells are not independent, they are from the exactly same stock split in 6, they are at the same cell density, they are given the same dilution of your nanoparticles etc. A small error here, would produce a certain phenotype. Repeating the experiment three times would eliminate small errors/fluctuations and produce a robust phenotype of your nanoformulation only. When I tried to find a reference for this I came across this paper: 

What exactly is ‘N’ in cell culture and animal experiments?

Stanley E. Lazic, 2018

First they state (which is in accordance with your view): "A well, culture dish, or another plastic container is the appropriate EU (experimental unit = biological entitiy) for cell culture experiments." 

But then it is emphasized (in accordance with our procedure in cell culture experiments): "But in vitro experiments are often finicky; the results depend on the unique conditions that vary each time the experiment is run. The experimental material (e.g., a cell line) is often artificially homogeneous and the conditions under which the experiment is run are so narrowly defined that it is hard to know what will happen if the experiment is run a second time. For this reason, in vitro experiments are usually repeated on multiple days, and the number of wells, aliquots, or culture dishes within a day are treated as subsamples. The aim is to establish that the phenomenon is robust enough to survive multiple replications of the entire experimental run or protocol in a highly artificial system. This situation has parallels to the nonhuman primate example above. We could do the experiment on 1 day and use the wells as the EU, and have a large sample size, but then we cannot comment on the generalisability of the results. If the experimental system is sensitive to the many details of how it is carried out, then repeating the whole procedure on multiple days provides further information in a way that using more wells on a single day does not. It provides an estimate of the consistency of the effects across the different experimental runs (days). The multiple wells on each day are then treated as subsamples and do not contribute to N (for example, by averaging values across wells in the same condition on each day). This is a scientific judgement about the relevant unit that we would like to make inferences about, and although opinions may differ, using more stringent criteria makes the results more believable."

Thus, when writing n = 6 in all figures is strongly misleading for cell biologists. We assume this is 6 independent experiments. Writing 6 treatments is slightly better, but the full truth is only revealed if you write: The graph shows the mean +- SD from one experiment performed with 6 replicate wells. 

Thus, I find it hard to accept the corrections performed on this paper, without changing the wording in each figure legend. I.e. to explicitly write that this is based on one experiment with 6 replicates, to expose the scientific foundation for the conclusions and the statistics. Then every reader can decide for themselves whether the data are trustworthy or not.
